# SAMPLE-EFFICIENT REINFORCEMENT LEARNING BY BREAKING THE REPLAY RATIO BARRIER

**Pierluca D'Oro***
Mila, Université de Montréal

**Max Schwarzer***
Google Brain
Mila, Université de Montréal

**Evgenii Nikishin**
Mila, Université de Montréal

**Pierre-Luc Bacon**
Mila, Université de Montréal

**Marc G. Bellemare**
Google Brain, Mila

**Aaron Courville**
Mila, Université de Montréal

## ABSTRACT

Increasing the replay ratio, the number of updates of an agent's parameters per environment interaction, is an appealing strategy for improving the sample efficiency of deep reinforcement learning algorithms. In this work, we show that fully or partially resetting the parameters of deep reinforcement learning agents causes better replay ratio scaling capabilities to emerge. We push the limits of the sample efficiency of carefully-modified algorithms by training them using an order of magnitude more updates than usual, significantly improving their performance in the Atari 100k and DeepMind Control Suite benchmarks. We then provide an analysis of the design choices required for favorable replay ratio scaling to be possible and discuss inherent limits and tradeoffs.

## 1 INTRODUCTION

In many real world scenarios, each interaction with the environment comes at a cost, and it is desirable for deep reinforcement learning (RL) algorithms to learn with a minimal amount of samples (François-Lavet et al., 2018). This can be naturally achieved if an algorithm is able to leverage more computational resources during training to improve its performance. Given the online nature of deep RL, there is a peculiar way to aim at having such behavior: to train the agent for longer, given a dataset of experiences, before interacting with the environment again.

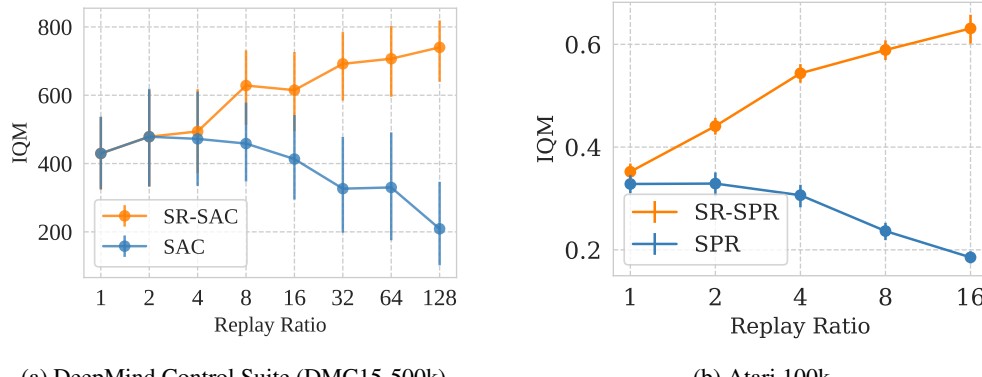

(a) DeepMind Control Suite (DMC15-500k)  (b) Atari 100k

Figure 1: Scaling behavior of SAC and SR-SAC in the DeepMind Control Suite (DMC15-500k) benchmark, and of SPR and SR-SPR in the Atari 100k benchmark (5 seeds for point for SAC and SR-SAC, at least 20 seeds for point for SPR and SR-SPR, 95% bootstrapped C.I.).

*Equal contribution. Correspondance to {pierluca.doro, schwarzm}@mila.quebec.

A method based on this idea can be said to be *scaling the replay ratio*, the number of updates of an agent's parameters for each environment interaction. Despite generally providing limited benefit when applied to standard baselines (Fedus et al., 2020; Kumar et al., 2021), replay ratio scaling has been shown to bring performance improvements to well-tuned algorithms. Recent approaches were able to achieve better sample efficiency by increasing it to higher values, up to $8$ for discrete control (Kielak, 2019) or 20 for continuous control (Chen et al., 2021; Smith et al., 2022).

In this paper, we show that it is possible, with minimal but careful modifications to model-free algorithms mostly based on parameter resets (Ash & Adams, 2020; Nikishin et al., 2022), to reach new levels of replay ratio scaling and push the sample efficiency limits of deep RL. Both in continuous control, with SAC in DeepMind Control Suite (Haarnoja et al., 2018; Tassa et al., 2018), and discrete control, with SPR in Atari 100k (Schwarzer et al., 2021a; Kaiser et al., 2020), we *break the replay ratio barrier*, unlocking a training regime in which orders of magnitude of additional agent updates can be used to increase the performance of an algorithm for a given budget of interactions with the environment. By doing so, we obtain better aggregated scores than strong baselines, with a general blueprint to improve sample efficiency of potentially any off-policy deep RL algorithm.

To understand how this can be feasible, it is useful to reflect on one of the most common patterns observed in the development of deep RL algorithms (Mnih et al., 2015b). With a few exceptions, researchers typically ground their methods on the well-established dynamic programming mathematical machinery, combining it with optimization strategies common in deep learning. However, the RL setting is inherently different from the one in which most deep learning architectures and optimization methods were developed. In deep RL, neural networks have to deal with dynamic datasets, whose composition changes over the course of training; their training actively determines the value of future inputs, but also the value of future targets. We argue that the recently identified tendency of neural networks to lose their ability to learn and generalize from new information during training (Chaudhry et al., 2018; Ash & Adams, 2020; Berariu et al., 2021; Igl et al., 2021; Dohare et al., 2022; Lyle et al., 2022a;b; Nikishin et al., 2022), against which most RL methods deploy no countermeasures, has been the main roadblock in achieving better sample efficiency through replay ratio scaling.

After presenting and evaluating our algorithmic solution leading to better replay ratio scaling, we discuss some of the aspects of thinking about deep RL algorithms under the lens of this paradigm. We show some examples of algorithm design decisions important, or not important, for effective replay ratio scaling to be possible, with particular attention to the role of online interaction. Then, we visualize in an explicit way the data-computations tradeoff implied by this approach and, after having shown the potential of replay ratio scaling, we discuss its inherent limits.

## 2  RELATED WORK

**Loss of Ability to Learn and Generalize in Neural Networks**   A growing body of evidence suggests that artificial neural networks lose their ability to learn and generalize during training. The phenomenon is not clearly visible when learning with a static dataset on a fixed task, but it starts appearing when the data distribution changes. In the continual learning setting, an alleviation of the problem by partially resetting the network parameters already provides a consistent improvement (Ash & Adams, 2020). Berariu et al. (2021) provides an in-depth study of how this phenomenon happens, including how many training updates are required for the performance of a network on future tasks to be unrecoverably damaged. The phenomenon becomes even more prominent in deep RL, where it has been identified in multiple settings. In the context of on-policy algorithms, it has been investigated as a consequence of *transient non-stationarity* and mitigated via self-distillation (Igl et al., 2021); in off-policy RL, it has been studied under the name of *capacity loss* (Lyle et al., 2022a), counteracted by the use of auxiliary tasks; in the sparse reward setting, it has been mitigated by post-training policy distillation (Lyle et al., 2022b). To address what they call *loss of plasticity*, Dohare et al. (2022) proposes a variation of backpropagation compatible with continual learning, also applying it to the continual RL context. In this paper, we primarily leverage a periodic hard resetting method (Zhou et al., 2022), as investigated in Nikishin et al. (2022) to address *the primacy bias* phenomenon. Our work demonstrates that addressing this phenomenon allows for increased sample efficiency by scaling the replay ratio to much higher values than other model-free methods. We report in Appendix A a more precise summary and glossary of the different related definitions from previous work.

**Scaling in Deep and Reinforcement Learning**   The topic of understanding and exploiting the scaling behavior of a deep learning algorithm's performance with respect to the amount of resources used for training has recently gained attention. Hestness et al. (2017) pioneered the idea of empirically studying and predicting performance when increasing a model's size, and subsequent work investigated scaling with respect to both model and dataset size, as well as training time (Kaplan et al., 2020; Bahri et al., 2021; Djolonga et al., 2021). Recent work in language modeling has also highlighted the importance of having high-quality data and the right training setup for efficient scaling to be possible (Hoffmann et al., 2022). In RL, scaling with respect to model size has been investigated in the offline setting for decision transformers (Lee et al., 2022) and with respect to planning-time in model-based RL (Hamrick et al., 2021). For what concerns replay ratio scaling, moderately increasing the replay ratio for standard baselines has been shown to be a competitive data-efficient baseline for both discrete and continuous control when compared to model-based RL methods (Holland et al., 2018; Van Hasselt et al., 2019; Kielak, 2019; D'Oro & Jaśkowski, 2020), despite clear limitations (Kumar et al., 2021). Recent approaches in continuous control leveraged high replay ratios as a strategy to improve sample efficiency through the use of ensembles of value functions (Chen et al., 2021; Hiraoka et al., 2022; Wu et al., 2022) or normalization strategies (Smith et al., 2022); we argue that explicitly alleviating the progressive loss of ability to learn and generalize pushes the replay ratio scaling capabilities much further than those techniques can achieve.

## 3   Effective Replay Ratio Scaling with Resets

Most off-policy deep RL algorithms make use of a replay buffer (Lin, 1992) for storing transitions encountered over (a window of) an agent's lifespan. At a fixed frequency, such methods sample a batch of transitions from the buffer, update the parameters of the agent by following the gradient of some loss function, and let the agent interact again with the environment before adding new experience to the buffer. The number of agent updates per environment step is usually called *replay ratio*[1] (Wang et al., 2016; Fedus et al., 2020), and most standard algorithms are trained with a value around 1 (Mnih et al., 2015a; Haarnoja et al., 2018). It is natural to view increasing the replay ratio beyond these values as a way to improve sample efficiency. For ease of discussion, we now explicitly state and give a name to this idea, which has been an object of interest in previous studies (Van Hasselt et al., 2019; Kumar et al., 2021).

> **Replay Ratio Scaling**
>
> Change in an agent's performance caused by doing more updates for a fixed number of environment interactions.

This definition does not have any positive connotation per se; any deep RL algorithm will have a certain replay ratio scaling behavior, and a desirable property for an algorithm is to have particularly *favorable* replay ratio scaling, so that its performance can improve by increasing the replay ratio.

In contrast to other performance scaling properties analyzed for deep learning algorithms (Kaplan et al., 2020), replay ratio scaling is intertwined with the online RL paradigm: if the agent has a significantly better data-collection policy due to more training, the next collected sample will be potentially different with respect to the one collected if doing less training before the interaction; by this virtue, also future learning will be directly impacted by the presence of different data in the replay buffer. In other words, this type of scaling can only be understood by considering the interaction of an agent with an environment: training more on a small dataset of interactions, without any further collection of data, will eventually lead to challenges associated to off-policy learning (Ostrovski et al., 2021); but training more *while the data is collected* can drastically change the stream of incoming data and the overall learning dynamics.

Given its appeal, what are the limiting factors to increasing the replay ratio? We argue that the main factor inhibiting effective replay ratio scaling in existing deep RL algorithms has been *the progressive loss of the ability to learn and generalize in neural networks* (Dohare et al., 2022; Lyle et al., 2022b; Nikishin et al., 2022). It has been shown that this property hinders a neural network's performance under task switches (Ash & Adams, 2020; Berariu et al., 2021) and, from the perspective

---

[1]Related quantities are also known as update-to-data (UTD) ratio (Chen et al., 2021; Smith et al., 2022).

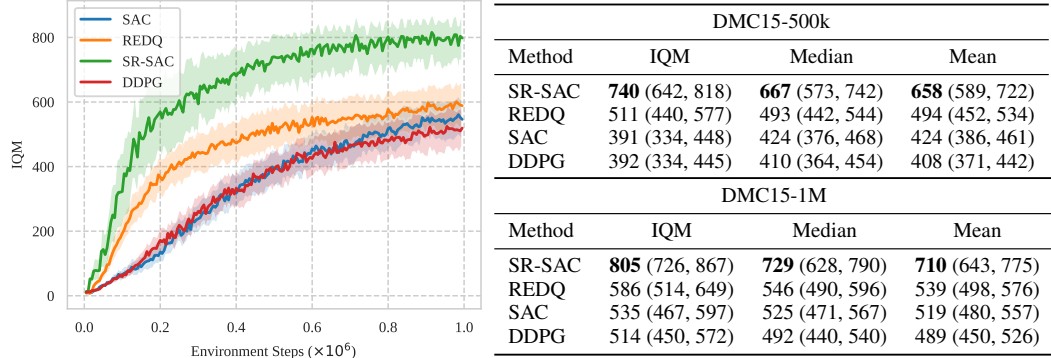

| | DMC15-500k | | |
| --- | --- | --- | --- |
| Method | IQM | Median | Mean |
| SR-SAC | **740** (642, 818) | **667** (573, 742) | **658** (589, 722) |
| REDQ | 511 (440, 577) | 493 (442, 544) | 494 (452, 534) |
| SAC | 391 (334, 448) | 424 (376, 468) | 424 (386, 461) |
| DDPG | 392 (334, 445) | 410 (364, 454) | 408 (371, 442) |
| | DMC15-1M | | |
| Method | IQM | Median | Mean |
| SR-SAC | **805** (726, 867) | **729** (628, 790) | **710** (643, 775) |
| REDQ | 586 (514, 649) | 546 (490, 596) | 539 (498, 576) |
| SAC | 535 (467, 597) | 525 (471, 567) | 519 (480, 557) |
| DDPG | 514 (450, 572) | 492 (440, 540) | 489 (450, 526) |

Figure 2 & Table 1: Performance of SR-SAC and of standard baselines on the DMC15 benchmark. (5 seeds for SR-SAC, 20 for all other algorithms, 95% bootstrapped C.I.).

of a neural network employed by the agent, what is deep RL if not a long sequence of related but distinct tasks (Dabney et al., 2021)?

Recent studies showed that, even under smooth task changes, the more training has been done on a previous task, the worse the performance will eventually be in a new task (Ash & Adams, 2020; Berariu et al., 2021). Since higher replay ratio correspond to an increased amount of training, this gives a natural explanation to the limit in increasing it. The ability to learn and generalize can, however, be restored. For instance, Nikishin et al. (2022) periodically reset the network's parameters, with a frequency that is fixed with respect to the number of *environment steps*. In this work, we argue that the key to surprisingly effective replay ratio scaling is a periodic restoration of the ability to learn and generalize of the network, via partial (Ash & Adams, 2020) or total (Nikishin et al., 2022) resets of its parameters, with a reset frequency that only depends on the number of *updates* and thus implicitly also on the replay ratio. This means the more an algorithm updates its neural networks, the more frequent the restoration of its ability to learn and generalize will be, leading to better performance, as we now show in practice.

## 4    REPLAY RATIO SCALING DRASTICALLY IMPROVES SAMPLE EFFICIENCY

We apply two different reset strategies to two standard continuous control and discrete control algorithms and study their replay ratio scaling behavior. We consider Soft Actor-Critic (SAC) (Haarnoja et al., 2018), which optimizes an actor and a critic by maximizing policy entropy alongside the environment's reward, and SPR (Schwarzer et al., 2021a), a model-free DQN-based reinforcement learning algorithm that augments a sample-efficient variant of Rainbow (Van Hasselt et al., 2019) with a model-based latent dynamics prediction objective designed to improve representation learning in the low-data regime. The two curves in Figure 1 show that it is possible, with the same algorithm, to almost double the performance for the same number of environment steps, by just varying the replay ratio. We call the modified versions of these two algorithms *Scaled-by-Resetting SAC* (SR-SAC) and *Scaled-by-Resetting SPR* (SR-SPR). In the rest of this section, we are going to describe the precise the details of the reset strategies that we employ for the two algorithms, as well as the benchmarks to which they are applied, by describing our decisions first in continuous control and then in discrete control. For evaluation and comparisons, we follow the protocol suggested by Agarwal et al. (2021).

### 4.1    CONTINUOUS CONTROL

**The DMC15 Benchmark**    To appropriately compare the performance of different algorithms, we consider a benchmark based on 15 environments from DeepMind Control Suite (Tassa et al., 2018). Our selection of tasks, reported in Table 6, is a set for which discussing sample efficiency is sensible (i.e., neither immediately solvable nor unsolvable by common deep RL algorithms). For ease of comparison, we specialize the benchmark to DMC15-500k, in which $5 \times 10^5$ interactions with the environment are allowed, and DMC15-1M, in which $10^6$ interactions are allowed.

**Reset Strategy** We adapt the approach of Nikishin et al. (2022), and completely reset all the agent parameters every $2.56 \times 10^6$ of its updates. This lets us avoid individually specifying the moments at which resets should happen for different replay ratios. In terms of environment steps, resets will just occur more often at higher replay ratios. For instance, for replay ratio 128 (128x higher than what typically used by SAC), a reset occurs once every 20000 steps of interaction with the environment.

**Results** In Figure 2, we compare a version of SR-SAC that uses a replay ratio of 128 to standard deep RL baselines. This also includes the recently proposed REDQ (Chen et al., 2021), which obtained state-of-the-art sample efficiency by using a replay ratio of 20. At any budget of interactions with the environment, SR-SAC compares favorably with REDQ, despite being a simpler algorithm. SR-SAC establishes a new state-of-the-art result for model-free continuous control. Following (Agarwal et al., 2021) we focus on interquartile mean (IQM) performance, defined as the 25% trimmed mean performance over all runs on all considered tasks, and report 95% bootstrap confidence intervals.

## 4.2 ATARI 100K

**Reset Strategy** We follow Nikishin et al. (2022) in performing one reset every 40,000 updates; at replay ratio 16, the highest considered, this corresponds to a reset every 2,500 environment steps, or roughly once every three minutes of interaction. However, Nikishin et al. (2022) only reset a subset of the agent's parameters when training on the ALE, leaving the agent's convolutional encoder untouched by resets. While this leaves the encoder vulnerable to plasticity loss, fully resetting the encoder is impractical, as Nikishin et al. (2022) observe. As an intermediate solution, we apply soft resets, using a variant of *Shrink and Perturb* (Ash & Adams, 2020) in which encoder parameters are interpolated between their previous value and a random re-initialized parameter vector on each reset: $\theta^t = \alpha\theta^{t-1} + (1-\alpha)\phi, \phi \sim$ `initializer`. This formulation is different from that used by (Ash & Adams, 2020) but allows easy interpolation between completely resetting a layer and leaving it unchanged; we use $\alpha = 0.8$ by default. We examine the impact of this decision in Section 5.2.

**Target Networks** By default, SPR does not employ a separate target network, unlike traditional DQNs (Mnih et al., 2015a). However, we find that this leads replay ratio scaling to stop improving performance at relatively low replay ratios, which we hypothesize is due to fundamental variance in optimization limiting the accuracy to which the value function may be estimated. To alleviate it, we directly adopt the target strategy employed by SR-SAC, with an exponential moving average (EMA) target network with coefficient $\tau = 0.005$, which we find allows beneficial replay ratio scaling out to at least replay ratio 16. Moreover, following (Ghavamzadeh et al., 2011), SR-SPR also uses its target network for action selection. We elaborate on this design decision in Section 5.2.

**Results** Figure 3 shows performance profiles of SR-SPR at various replay ratios, demonstrating that replay ratio scaling consistently improves performance up to at least replay ratio 16. We also compare a version of SR-SPR that uses replay ratio 16 to standard baselines (DrQ, DER, Kostrikov

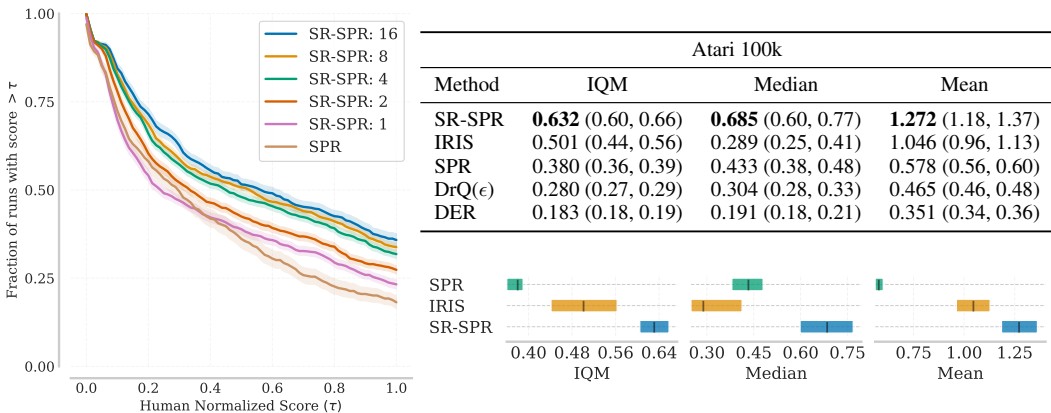

| | Atari 100k | | |
|---|---|---|---|
| Method | IQM | Median | Mean |
| SR-SPR | **0.632** (0.60, 0.66) | **0.685** (0.60, 0.77) | **1.272** (1.18, 1.37) |
| IRIS | 0.501 (0.44, 0.56) | 0.289 (0.25, 0.41) | 1.046 (0.96, 1.13) |
| SPR | 0.380 (0.36, 0.39) | 0.433 (0.38, 0.48) | 0.578 (0.56, 0.60) |
| DrQ($\epsilon$) | 0.280 (0.27, 0.29) | 0.304 (0.28, 0.33) | 0.465 (0.46, 0.48) |
| DER | 0.183 (0.18, 0.19) | 0.191 (0.18, 0.21) | 0.351 (0.34, 0.36) |

Figure 3 & Table 2: Performance profiles (left, higher is better) of SR-SPR at various replay ratios, and 95% C.I.s of SR-SPR: 16 and of standard baselines on Atari 100k (right, 20 seeds for SR-SPR and SPR, 5 seeds for IRIS, 100 seeds for all other algorithms as taken from Agarwal et al. (2021))

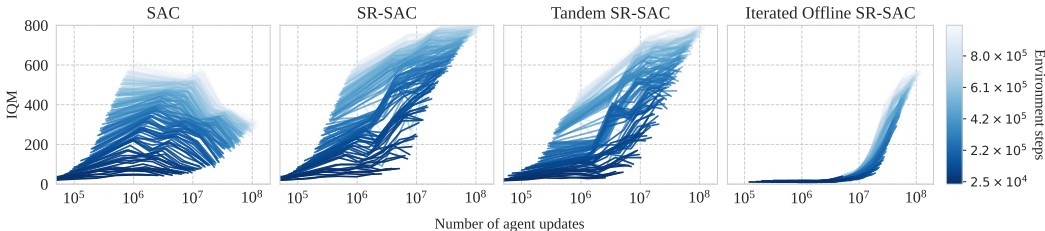

Figure 4: Scaling behavior of SAC, SR-SAC and its tandem and iterated offline variations in the DMC15 benchmark. Each individual line shows performance at a given number of environment steps, denoted by color, across different numbers of agent updates. Each point in a line is obtained by measuring performance with a different replay ratio for that number of environment steps. Each line is computed over 5 seeds.

et al., 2022; Van Hasselt et al., 2019) and recent work (IRIS, Micheli et al., 2022) in table 2. SR-SPR establishes a new state-of-the-art for model-free control on Atari 100k, and rivals prior work that has aggressively pretrained on additional data (Liu & Abbeel, 2021; Schwarzer et al., 2021b). We present full results and per-game scores for SR-SPR in table 4, and show training curves in fig. 15. We report IQM performance, as well as plotting a performance profile (Agarwal et al., 2021), which visualizes the full distribution of performance across all runs[2] and demonstrates that increasing SR-SPR's replay ratio comprehensively improves performance.

# 5 ALGORITHM DESIGN IN LIGHT OF REPLAY RATIO SCALING

## 5.1 ANALYZING THE IMPORTANCE OF ONLINE INTERACTION

When training with high replay ratios and short reset intervals, the training regime an agent is subjected to begins to resemble offline RL; the agent is primarily learning from data collected by policies unrelated to its own, with only a small amount of online data available to correct its policy. Given many classical analyses from offline RL (Levine et al., 2020), it is perhaps surprising that an agent trained in a pseudo-offline setting with no explicit regularization towards conservatism (e.g., Kumar et al., 2020) can learn successfully. What is then the role of the incoming stream of interactions? To gain some understanding, in this section we attack the problem from different angles and study the scaling behavior of variants of SR-SAC. We consider different data collection patterns and how interleaving them with agent optimization changes the training dynamics. The appendix also presents a comparisons with NFQI (Riedmiller, 2005) and with the online use of an offline RL algorithm.

### 5.1.1 ITERATED OFFLINE SETTING

Changing the replay ratio in a deep RL algorithm can be seen as a specific way of increasing the proportion of offline training an agent is subject to. Specifically, the agent's parameters are updated a number of times exactly equal to the replay ratio before a new sample is collected. This implies a uniform distribution of the number of offline updates across time steps. Is this an important variable for determining the replay ratio scaling behavior of an algorithm?

To answer this question, we resort to what we call *iterated offline* RL (Matsushima et al., 2021; Riedmiller et al., 2021), which alternates between purely offline updates and data collection. In this setting, a certain value of replay ratio is not distributed uniformly during the course of the interactions with the environment. Instead, the agent is not updated during data collection, and an amount of updates equal to the one that would be due in that data collection time frame in virtue of the replay ratio is applied completely offline, right after each reset.

As visible in Figure 4, the iterated offline paradigm has a different replay ratio scaling behavior. Applying a very large number of updates with a fixed dataset, with an algorithm such as SAC, incurs serious risk of generating a degenerate policy, not able to outperform the previous one. As exemplified in Figure 6, in the absence of any mechanism for stopping this natural degeneration to happen, this circle is broken only when enough data is collected. Collecting enough data in this sense is possible

---

[2]Broadly speaking, a transposed and clipped plot of the cumulative distribution function of performance.

Figure 6: Examples of behaviors of SR-SAC and its tandem and iterated offline variations on four environments from DMC15. (5 runs, ± std).

for easy tasks such as `hopper-stand` and `walker-run`, with a cost in sample efficiency, or impossible on hard tasks such as `humanoid-stand` and `quadruped-walk`. This is unfortunate: the iterated offline RL paradigm can be quite useful in practical settings, in which the agent is allowed to only collected batches of data without any update (perhaps for safety reasons); however, current backbone algorithms (such as SAC) are not currently compatible with such a setting, that thus leads to favorable replay scaling only when closer to the online setting. This explains the sudden increase in the curve of Figure 4, when the number of agent updates, and consequently the reset frequency, becomes large enough. An interesting question, left for future work, is whether this behavior could change if combined with conservative algorithms created for the offline RL setting.

### 5.1.2 TANDEM SETTING

With high replay ratios, an agent's training begins to resemble offline RL: although the agent still has the possibility to interact with the environment, it is very infrequent relative to the amount of training. Thus, an agent after a reset has a small stream of interactions collected by the agent itself, while the vast majority of its data was collected by potentially unrelated agents. How important, then, is this small stream of online interaction? To answer this question, we leverage the tandem setting, as presented in Ostrovski et al. (2021). Two copies of the same agent, identical apart from the initialization, are created. With the same algorithm (SR-SAC in this case), they are trained on the replay buffer collected by the *active agent*. The *passive agent* thus never directly interacts with environment, and cannot collect data to correct its own misconceptions about the environment.

As shown in Figure 4, the behavior of Tandem SR-SAC offers an alternative perspective on the importance of online interactions: despite the performance of the algorithm being hurt, the overall replay ratio scaling capabilities remain similar. We can look at the performance of the passive agent to understand what the exact effect of online interactions is on training. As evident in the environments from Figure 6, especially in `hopper-stand` and `quadruped-walk`, there is a qualitative difference between the behavior of an active agent (blue curve) and a passive agent (green curve): right after a reset, with the initial high replay ratio training, the performance of both agents is greatly improved; after a few thousands steps, training remains stable for the active agent but causes performance collapse in the passive agent. This experiment thus demonstrates the power of having online interactions as an *implicit regularization mechanism*.

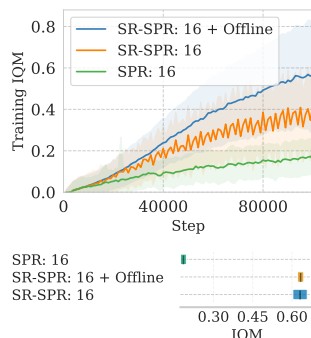

Figure 5: Learning curves (top) and evaluation performance (bottom) at replay ratio 16 for SPR and SR-SPR with and without offline updates after each reset.

For the design of future replay ratio-scalable algorithms, one should keep in mind that it is indeed possible to scale an algorithm potentially affected by extreme off-policyness; however, online data collection slows down performance collapse when training aggressively, as shown in both the iterated offline and the tandem experiments.

### 5.1.3 ALTERNATIVE COMBINATIONS OF OFFLINE AND ONLINE UPDATES

The iterated offline setting can be seen as the extreme in which all of the updates are done offline, compared to the even distribution used in the online setting. What if we use an intermediate strategy?

For SR-SPR, we find that directly mixing offline and online RL by performing half the updates allotted to each interval immediately after each reset can actually improve performance by some metrics, such as training return (see Figure 5 upper), by mitigating the performance drop otherwise experienced after each reset. Although we find that this has essentially no impact on final evaluation performance (Figure 5 lower), it may allow SR-SPR to be used when cumulative regret is important.

## 5.2 WHAT IS REQUIRED FOR REPLAY RATIO SCALING IN DISCRETE CONTROL?

Although replay ratio scaling is relatively straightforward for SR-SAC, achieving robust replay ratio scaling for SR-SPR requires more complex design decisions due to its shorter training period and more complex function approximation. As a result, unlike SR-SAC, SR-SPR contains additional modifications from the variant of SPR used by Nikishin et al. (2022). We study the impact of these design decisions on scaling behavior and report results in Figure 7.

Inspired by the findings of Berariu et al. (2021) that plasticity loss is concentrated in the final layers of the network but affects all layers, we apply Shrink and Perturb (SP) to the encoder; this is responsible for roughly a constant increase of IQM 0.04 past replay ratio 4. We note however that applying Shrink and Perturb alone to all the parameters of the network is not sufficient to enable beneficial scaling; it is important that at least the network's final layers be completely reset. We explain this using the observations from (Berariu et al., 2021) that the last layers are more responsible for the loss of plasticity.

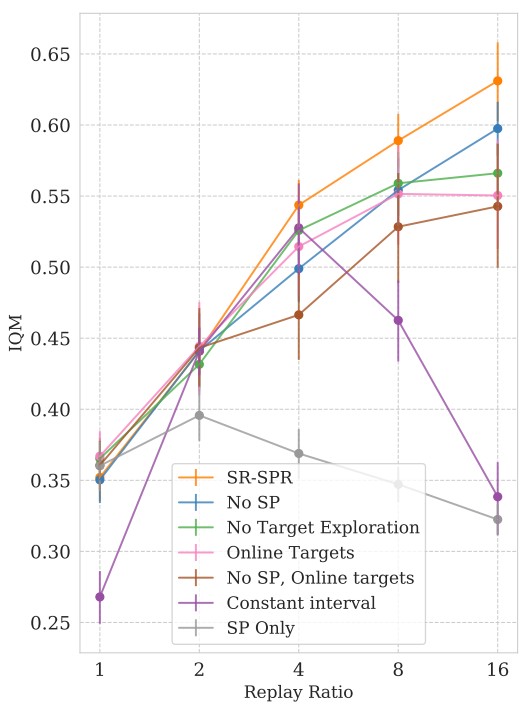

Figure 7: The replay ratio scaling behavior of SR-SPR with various components ablated.

That said, the most important factor in allowing SR-SPR to continue scaling well is its use of a target network. This effect is primarily due to better action selection through the target network; we found that the stabilizing effect on optimization was a less important factor. This is reminiscent of speedy Q-learning (Ghavamzadeh et al., 2011), where the use of an exponential moving average policy was shown to improve convergence speed, and can also be understood in relationship to the policy churn phenomenon (Schaul et al., 2022) (see Figure 10 in the appendix).

Meanwhile, removing both Shrink and Perturb and the target network is roughly equivalent to taking the method of Nikishin et al. (2022) but setting reset intervals as in SR-SPR. As Figure 7 suggests, this alone suffices to yield some replay ratio scaling but not as efficient compared to SR-SPR. However, maintaining a fixed reset interval (in terms of environment steps) when varying replay ratio, as done by Nikishin et al. (2022), leads to poor performance at replay ratios above 4.

Intriguingly, we note that these modifications are beneficial specifically for replay ratio scaling; at replay ratios 1 or 2 they do not improve performance (although for the most part they do not significantly harm performance either). We thus hypothesize that there may be other modifications to complex algorithms such as SPR that could be made to further improve replay ratio scaling properties, but that are today not in widespread use because they do not improve performance in standard low replay ratio settings.

## 5.3 VISUALIZING THE DATA/COMPUTE TRADEOFF

If an order of magnitude more of updates can be used for improving the performance of an algorithm, additional tradeoffs start to emerge. The type of computations that replay ratio scaling implies are fundamentally different than other concepts of scaling, (e.g., about larger models): scaling here is inherently sequential. Thus, obtaining more hardware does not help faster execution of the algorithm.

When collecting new transitions is not very expensive, the choice between collecting new samples in the environment and spending more time updating an agent could become nontrivial.

We visualize this tradeoff in Figure 8. The plot is obtained by combining runs of SR-SAC with doubling replay ratio from 0.25 to 128, and considering, for a fixed data budget (in terms of environment steps), the total computational budget (in terms of total number of agent updates at that point), as well as the achieved performance. There exists multiple ways to achieve the same level of performance, as denoted by the color. This plot shows that resets provide a knob on replay ratio scaling and allows to tradeoff data for computation. If, for a given problem, sample efficiency is more important than computational considerations, one can spend about two orders of magnitude of additional agent updates to obtain the same performance that can be obtained by waiting for 800000 additional samples to be collected from the environment. The peculiar feature of the approach we advocate for in this paper is that it allows to act on this tradeoff with an algorithm basically as simple as the employed backbone.

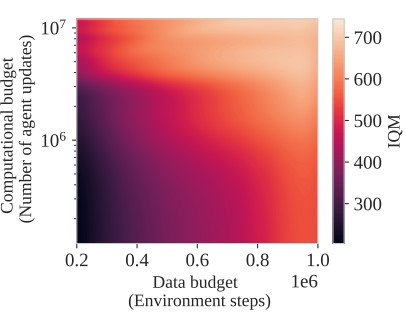

Figure 8: Performance of SR-SAC in DMC15 as a function of the number of interactions and of the number of agent updates, determined by the replay ratio.

## 6 THE LIMITS OF REPLAY RATIO SCALING

We have seen what becomes possible when higher level of replay ratio scaling are unlocked by resets. What are the limits of this paradigm? First of all, replay ratio scaling is always possible up to a finite value, at which there is simply not enough information left to be extracted from the existing dataset of experience. Current methods, including the one proposed in this paper, are not able to automatically identify when this limit is reached, and they are therefore still subject to performance collapse when increasing the replay ratio too much. Second, replay ratio scaling cannot go beyond the intrinsic limitations of the given deep RL algorithm: for example, if the task is simply impossible to solve because of hard credit assignment or exploration, then replay ratio scaling is only of limited help. Third, the strategy we proposed for replay ratio scaling is based on keeping the entire history of interactions with the environment in the replay buffer. While this is feasible for the kind of sample-efficiency benchmarks that we have used in this paper, it might also require special consideration to be applied to larger problems; for instance, it is possible to keep a large replay buffer on permanent storage, albeit at the cost of slower batch retrieval. Lastly, replay ratio scaling can inherently become time-consuming for a training agent, which can limit the applicability of methodologies like ours to settings requiring high-frequency interactions with an environment.

## 7 CONCLUSIONS

In this paper, we have shown that, by leveraging partial or full resets of an agent's parameters, it is possible to unlock new levels of favorable replay ratio scaling and, consequently, of sample-efficiency for model-free deep RL algorithms. We demonstrated this by a careful evaluation on the DeepMind Control Suite and Atari 100k benchmarks, where our approach (SR-SAC and SR-SPR) demonstrated far superior performance compared to strong baselines, with minimal amounts of additional algorithmic complexity. Then, we discussed which algorithmic design choices are important for achieving such levels of replay ratio scaling with a deep RL algorithm, as well as the tradeoffs implied by this paradigm. Through our empirical analysis, we showed the value of online data collection, offering a perspective on its relationship with offline RL (Levine et al., 2020).

More generally, this paper is about how to leverage a discovery for the design of future deep RL algorithms. We believe this work to be an example of how the development of effective deep RL methods should be achieved not only through extending existing algorithms or creating new ones, but also through the discovery of new phenomena related to deep RL systems, and of techniques for exploiting them to increase performance. It is natural to wonder whether deeper understanding or exploitation of surprising empirical properties (Ostrovski et al., 2021; Schaul et al., 2022) beyond the one behind this work could lead to the emergence of new capabilities in deep RL algorithms.

ACKNOWLEDGMENTS

The authors thank Zhixuan Lin for adapting the REDQ baseline, Nathan U. Rahn, Rishabh Agarwal, David Yu-Tung Hui, Jesse Farebrother for insightful discussions and useful suggestions on the early draft, the Mila community for creating a stimulating research environment, Digital Research Alliance of Canada and Nvidia for computational resources. This work was partially supported by CIFAR, Samsung, Hitachi, Facebook AI Chair, Borealis, IVADO, and Gruppo Ermenegildo Zegna.

We acknowledge the Python community (Van Rossum & Drake Jr, 1995; Oliphant, 2007) for developing the core set of tools that enabled this work, including JAX (Bradbury et al., 2018; Babuschkin et al., 2020), Jupyter (Kluyver et al., 2016), Matplotlib (Hunter, 2007), numpy (Oliphant, 2006; Van Der Walt et al., 2011), pandas (McKinney, 2012), and SciPy (Jones et al., 2014).

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

openreview.net/forum?id=ei3SY1_zYsE`.

| Expression | Definition | Used In |
|---|---|---|
| Damage from Warm-Starting | [Phenomenon for which] "a warm-started network performs worse on test samples than a network trained on the same data but with a new random initialization" | Ash & Adams (2020) |
| Damage from Non-Stationarity | "A memory effect where these transient non-stationarities can permanently impact the latent representation and adversely affect generalisation performance" | Igl et al. (2021) |
| Capacity Loss | "Reduced ability to fit new targets in deep neural networks" | Lyle et al. (2022b), Lyle et al. (2022a) |
| Loss of Plasticity | "Loss of the ability of the model to keep learning" | Berariu et al. (2021), Dohare et al. (2022) |
| Primacy Bias | "A tendency to overfit initial experiences that damages the rest of the learning process" | Nikishin et al. (2022) |

Table 3: Definitions of coinciding and related phenomena from previous work justifying the effectiveness of our strategy for replay ratio scaling.

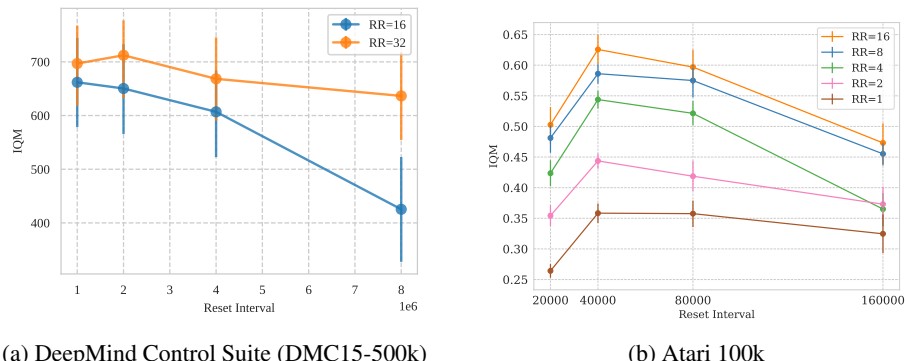

(a) DeepMind Control Suite (DMC15-500k)      (b) Atari 100k

Figure 9: Sensitivity of the IQM to varying reset intervals (in terms of gradient updates) of SR-SAC on the DeepMind Control Suite (DMC15-500k) benchmark, and of SR-SPR on the Atari 100k benchmark. (10 seeds, 95% bootstrapped C.I.).

# A  DEFINITIONS FROM RELATED WORKS

To further clarify our description of previous work from the related work section, we report in Table 3 definitions of the different terms used to refer to the loss of the ability to learn and generalize in neural networks. Each definition is directly taken from one of the papers corresponding to it. Note that, despite their overlap, they reflect slightly different perspectives on the nature of this phenomenon, and it can be worth for future investigations to pin down which one of these is more relevant for replay ratio scaling or reinforcement learning as a whole.

# B  ADDITIONAL EXPERIMENTAL RESULTS

## B.1  ADDITIONAL STUDIES

**Reset Interval**  An important hyperparameter for both SR-SAC and SR-SPR is the interval at which resets are performed, as denominated in terms of number of agent updates. In Figure 9, we study how performance is impacted by this choice, at different replay ratios. Overall, both SR-SAC and SR-SPR perform well for a vast range of reset intervals, with favorable replay ratio scaling and generally smooth performance degradation. Note that, for large intervals (e.g., the last point on the right for SR-SAC with $RR = 16$, and last two points in the bottom right for SR-SPR), this is equivalent to actually performing no resets, just running the unmodified baseline algorithms. Thus, performance experiences non-smooth drops only in these easily avoidable cases.

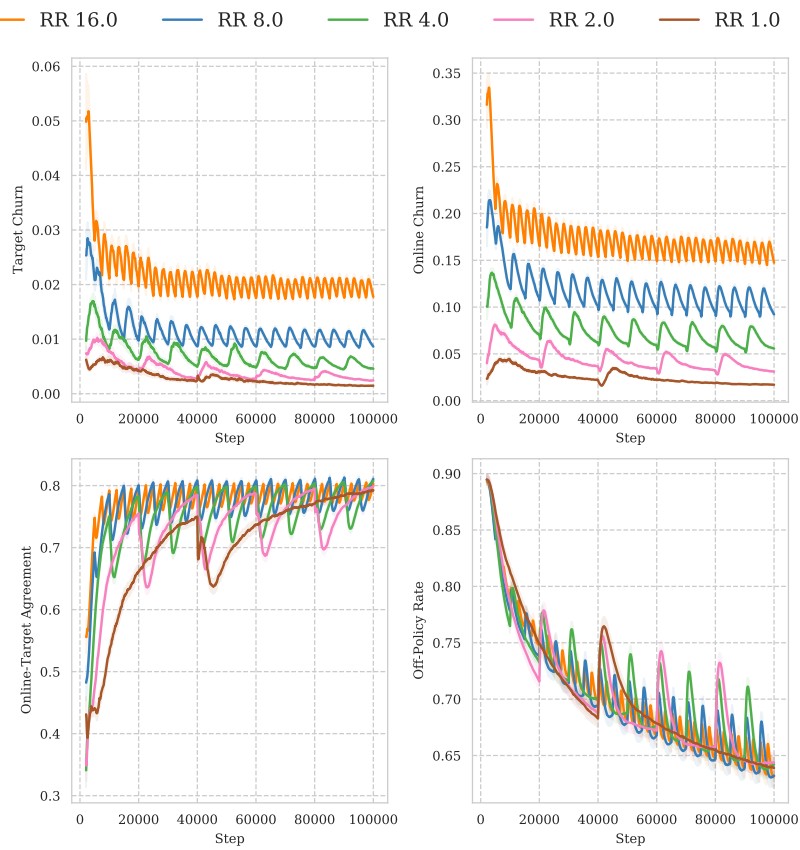

Figure 10: Churn-related diagnostics (based on the policy churn definition from Schaul et al. (2022)) for the online and target networks. Different colors and RR denote different values of replay ratio.

**Counteracting Policy Churn by Acting with the Target Network**    Schaul et al. (2022) defined the policy churn as the change in the agent's policy due to optimization. It was shown that a certain amount of policy churn can be beneficial for exploration in the absence of external noise (e.g., coming from $\epsilon$-greedy exploration); however, it is intuitive that excessive churn can actually hurt performance, for instance by breaking the the consistency of trajectories. We hypothesize that mitigating excessive policy churn is a major reason why SR-SPR performs better when actions are selected with the target network rather than the online network. Figure 10 shows that, if we measure churn before each interaction with the environment, increasing the replay ratio will naturally increase it, while acting with the target network will decrease it. When the replay ratio is too high, it is likely that the benefits coming from additional offline computations might be nullified by the inconsistency in the exploration data; acting with the target network reduces these inconsistencies without giving up on the more efficient optimization, at the cost of introducing minimal delays in the improvement of the data-collecting policy.

**Using an offline RL algorithm online**    In Section 5.1, we conducted a set of experiments with the goal of highlighting the importance of online interactions, concluding that a consistent stream of online interaction data and neural networks able to learn and generalize from a dataset of experiences are key factors behind effective replay ratio scaling. Online RL algorithms such as SAC are naturally reliant on the online stream of interactions; but it is natural to ask whether algorithms created for offline RL setting, where no interaction with the environment is assumed to be possible during training, can make the most out of the computational budget granted through high replay ratios in the online setting. To test this hypothesis, we run the Implicit Q-learning (IQL) (Kostrikov et al., 2022) algorithm as an online RL algorithm on DMC15-500k with a replay ratio of 32. Results in Figure 11 show that, despite being by design more robust to more aggressive training, the conservative

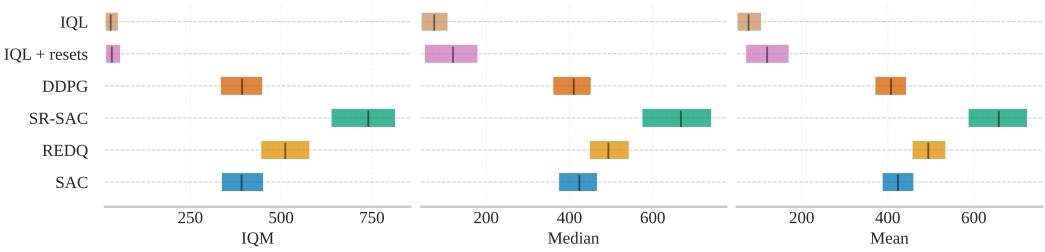

Figure 11: Performance on DMC15-500k of running IQL (RR=32) online, with and without resets.

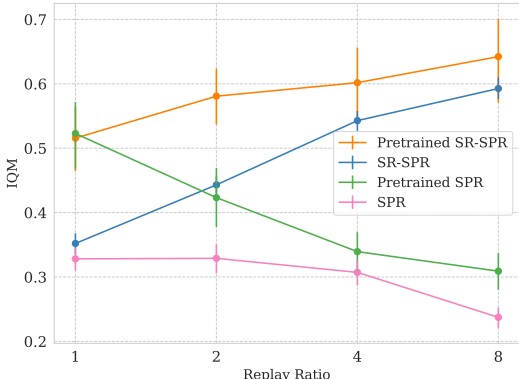

Figure 12: The performance of SR-SPR and SPR from scratch and when fine-tuning a pre-trained encoder on Atari 100k (10 seeds. 95% bootstrapped C.I.

nature of offline RL algorithms makes them not amenable to effective online learning, regardless of the presence of resets. This shows the effectiveness of online interactions as a strong supervision mechanism, able, when supported by resets, to make online RL algorithms robust to high replay ratios without the need of overly conservative behaviors.

## B.2 FINETUNING PRETRAINED REPRESENTATIONS

Finetuning pretrained representations has become increasing common in reinforcement learning and elsewhere (Schwarzer et al., 2021b; Liu & Abbeel, 2021; Bommasani et al., 2021). One obvious question to ask is whether or not replay scaling as demonstrated in the tabula rasa setting here can also be used to make this finetuning more sample efficient. In Figure 12 we answer this question in the affirmative. We initialize SPR and SR-SPR with pretrained encoders (taken for experimental convenience from SPR agents trained at replay ratio 1 for one million steps), and initialize all other parameters randomly. We then train at a range of replay ratios for 100k steps. For SR-SPR, we apply shrink and perturb towards the pretrained encoder weights rather than random parameters, but otherwise train as normal.

We find that while both SPR and SR-SPR benefit from the pretrained representations at low replay ratios, only SR-SPR is able to improve fine-tuning performance by replay scaling. Standard SPR with pretrained representations rapidly degrades in performance as the replay ratio is increased, while the performance of SR-SPR steadily increases at higher replay ratios. Although the gap between SR-SPR with and without pretraining closes somewhat at higher replay ratios, this is to be expected, as higher replay-ratio agents have more opportunities to improve their own representations even without pretraining.

## B.3 FINETUNING AFTER OFFLINE TRAINING

The efficiency of SR-SAC and SR-SPR makes the general approach behind their design potentially appealing for the setting of offline RL with an additional fine tuning phase. In Figure 13, we provide

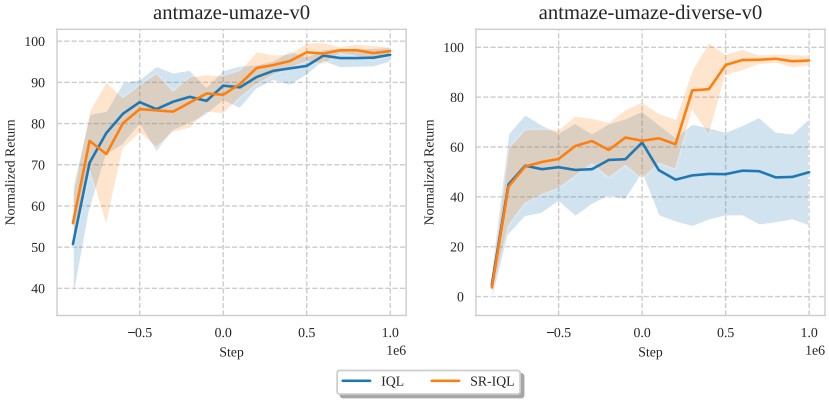

Figure 13: Performance of SR-IQL and IQL in two tasks from D4RL. Negative steps denotes the pretraining phase (10 seeds, $\pm$ std).

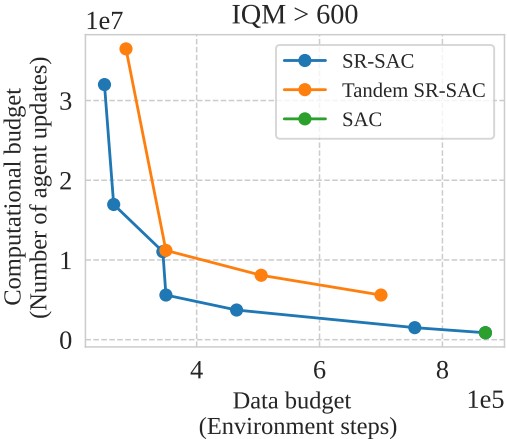

Figure 14: Pareto fronts for SR-SAC and Tandem SR-SAC on DMC15 (5 seeds).

preliminary evidence that the paradigm we advocated for in this paper may indeed be particularly beneficial in this setting. We test IQL with the same pretraining scheme presented in Kostrikov et al. (2022), consisting in a million offline training steps followed by a million interactions with the environment for fine tuning. We implement SR-IQL by using a replay ratio of 10 and resetting, every two million updates, all of the parameters of its neural networks during the fine tuning phase. We compare SR-IQL to IQL on two tasks from the D4RL benchmark (Fu et al., 2020). As shown in Figure 13, SR-IQL is roughly on par with IQL in the `antmaze-umaze-v0` task, but reaches superior performance during fine tuning in `antmaze-umaze-diverse-v0`. We believe this sets the stage to experimenting with our replay ratio scaling paradigm in this setting as a promising research direction for future work.

## B.4 PARETO FRONTS COMPARISON

With the same approach used for studying the data/computation tradeoffs of SR-SAC, it also becomes possible to directly compare the performance of different replay ratio-scalable algorithms. As a simple example, we compare SR-SAC, its tandem version and SAC in Figure 14. The different lines are Pareto curves, obtained by retaining the points that are dominating the other ones in terms of either data or computational budget, to reach an IQM of at least 600. On this plot, SAC simply appears as a point because, not allowing for effective replay ratio scaling, it can only reach the prescribed performance by using more data and a relatively small amount of computational resources.

### B.5 Comparison with Neural Fitted Q-Iteration

The approach we demonstrated for replay ratio scaling, for its relationship with offline RL and its use of resets, could resemble the classic NFQI algorithm (Riedmiller, 2005), which train from scratch, after each large batch of transitions, a Q-function. Our approach propagates information across resets mainly through the use of the replay buffer, having a fast target network updated alongside the regular agent training; NFQI instead propagates information primarily through a target network, which is updated once per reset. We implement an-friendly variant of NFQI on SR-SPR at replay ratio 16, performing one target network update upon each reset. However, we find that this leads to very poor performance (IQM 0.350), achieving barely half that of standard SR-SPR. Although we hypothesized that 2,500 environment steps (the standard reset interval for SR-SPR at replay ratio 16) might be too infrequent for target network updates, making this interval shorter did not improve performance. Although we cannot rule out the possibility that NFQI might be competitive at dramatically higher replay ratios, its inherent slowness in propagating information is likely to lead it to lag in data-efficient settings; any reset interval that is sufficiently long to allow for accurate estimation of the value function may lead to insufficiently rapid value propagation via target updates, and vice versa.

## C Computational Considerations

For DMC, the running time depends on the individual environment, due to differences in dimensionality of the observation as well as physics simulation time. On an NVIDIA V100 GPU, at this highest replay ratio of RR=128, our code takes about 10.5 hours on `acrobot-swingup` and about 15 hours on `humanoid-run` to complete 500k environment steps. For a replay ratio of RR=32, which yields remarkable, even if not best, performance, the time goes down to just about 3 hours and about 4 hours respectively. which is well-below the typical demands of modern model-based RL methods. With careful seed parallelization, running SR-SAC with RR=32 for 5 seeds for all tasks in the DMC15-500k benchmark takes less than 4 GPU/days on an NVIDIA V100. For Atari 100k, running time depends primarily on the replay ratio chosen. At the highest replay ratio used (16) and with five seeds running in parallel, our code takes roughly 25 hours to complete 100k steps on an NVIDIA A100, yielding a cost of roughly 5 GPU/hours per training run.

## D Experimental Details

We report in Table 6 the full list of tasks for the DMC15 benchmark. Our implementation of continuous control algorithms is based on the jaxrl codebase (Kostrikov, 2021). For REDQ, we use the best hyperparameters, as recommended by Chen et al. (2021), as well as a replay ratio of 20. For discrete control, we use a version of SPR implemented in Jax (Bradbury et al., 2018) in Dopamine (Castro et al., 2018). See Table 5 for a full list of the employed hyperparameters.

### D.1 Full Experimental Results

In Figure 16, we show the scaling curve for DMC15-1M.

We report in Table 4 the full per-game results for SR-SPR and in Figure 17,18,19,20,21 full experimental results for SR-SAC. For completeness, we also report the performance of the modified settings and of REDQ at the same replay ratios.

| Game | Random | Human | IRIS | SR-SPR:2 | SR-SPR:4 | SR-SPR:8 | SR-SPR:16 |
|---|---|---|---|---|---|---|---|
| Alien | 227.8 | 7127.7 | 420.0 | 877.9 | 964.4 | 1015.5 | **1107.8** |
| Amidar | 5.8 | 1719.5 | 143.0 | 189.2 | **211.8** | 203.1 | 203.4 |
| Assault | 222.4 | 742.0 | **1524.4** | 891.9 | 987.3 | 1069.5 | 1088.9 |
| Asterix | 210.0 | 8503.3 | 853.6 | 836.7 | 894.2 | **916.5** | 903.1 |
| Bank Heist | 14.2 | 753.1 | 53.1 | 253.6 | 460.0 | 472.3 | **531.7** |
| Battle Zone | 2360.0 | 37187.5 | 13074.0 | 14493.5 | 17800.6 | **19398.4** | 17671.0 |
| Boxing | 0.1 | 12.1 | **70.1** | 36.1 | 42.0 | 46.7 | 45.8 |
| Breakout | 1.7 | 30.5 | **83.7** | 24.5 | 26.1 | 28.8 | 25.5 |
| Chopper Command | 811.0 | 7387.8 | 1565.0 | 1609.4 | 1933.7 | 2201.0 | **2362.1** |
| Crazy Climber | 10780.5 | 35829.4 | **59324.2** | 28004.7 | 38341.7 | 43122.3 | 45544.1 |
| Demon Attack | 152.1 | 1971.0 | 2034.4 | 2969.0 | **3016.2** | 2898.1 | 2814.4 |
| Freeway | 0.0 | 29.6 | **31.1** | 24.1 | 24.5 | 24.9 | 25.4 |
| Frostbite | 65.2 | 4334.7 | 259.1 | 1450.4 | 1809.9 | 1752.8 | **2584.8** |
| Gopher | 257.6 | 2412.5 | **2236.1** | 735.3 | 717.5 | 711.2 | 712.4 |
| Hero | 1027.0 | 30826.4 | 7037.4 | 6832.1 | 7195.7 | 7679.6 | **8524.0** |
| Jamesbond | 29.0 | 302.8 | **462.7** | 412.9 | 408.8 | 392.8 | 389.1 |
| Kangaroo | 52.0 | 3035.0 | 838.2 | 1651.2 | 2024.1 | 3254.9 | **3631.7** |
| Krull | 1598.0 | 2665.5 | **6616.4** | 5206.4 | 5364.3 | 5824.8 | 5914.4 |
| Kung Fu Master | 258.5 | 22736.3 | **21759.8** | 14165.6 | 17656.5 | 17095.6 | 18649.4 |
| Ms Pacman | 307.3 | 6951.6 | 999.1 | 1472.6 | 1544.7 | 1522.6 | **1574.1** |
| Pong | -20.7 | 14.6 | **14.6** | -10.5 | -5.5 | -3.0 | 2.9 |
| Private Eye | 24.9 | 69571.3 | **100.0** | 98.8 | 95.8 | 95.8 | 97.9 |
| Qbert | 163.9 | 13455.0 | 745.7 | 3431.7 | 3699.8 | 3850.6 | **4044.1** |
| Road Runner | 11.5 | 7845.0 | 9614.6 | 12199.0 | **14287.3** | 13623.5 | 13463.4 |
| Seaquest | 68.4 | 42054.7 | 661.3 | 714.7 | 766.6 | 800.5 | **819.0** |
| Up N Down | 533.4 | 11693.2 | 3546.2 | 61851.2 | 91435.2 | 95501.1 | **112450.3** |
| Games > Human | 0 | 0 | **9** | 7 | 8 | **9** | **9** |
| IQM (↑) | 0.000 | 1.000 | 0.501 | 0.444 | 0.544 | 0.589 | **0.632** |
| Optimality Gap (↓) | 1.000 | 0.000 | 0.512 | 0.516 | 0.470 | 0.452 | **0.433** |
| Median (↑) | 0.000 | 1.000 | 0.289 | 0.336 | 0.523 | 0.560 | **0.685** |
| Mean (↑) | 0.000 | 1.000 | 1.046 | 0.910 | 1.111 | 1.188 | **1.272** |

Table 4: Full results for individual games in Atari 100k for SR-SPR at various replay ratios.

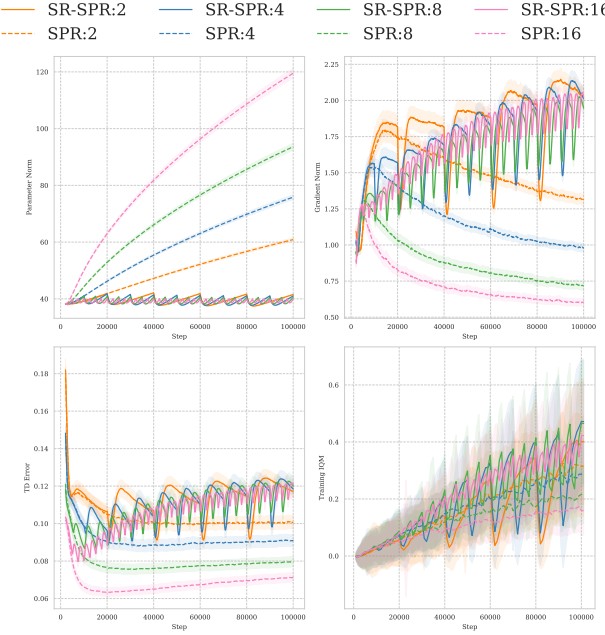

Figure 15: Learning curves for SR-SPR (solid) and SPR (dashed) at various replay ratios. Note that all SR-SPR runs converge to similar TD errors, gradient norms and parameter norms, while these metrics greatly differ for SPR at different replay ratios. IQM training performance does not match evaluation performance, as ongoing training episodes are often disrupted by the reset procedure.

| Parameter | Setting |
|---|---|
| Gray-scaling | True |
| Observation down-sampling | 84x84 |
| Frames stacked | 4 |
| Action repetitions | 4 |
| Reward clipping | [-1, 1] |
| Terminal on loss of life | True |
| Max frames per episode | 108K |
| Update | Distributional Q |
| Dueling | True |
| Support of Q-distribution | 51 |
| Discount factor | 0.99 |
| Minibatch size | 32 |
| Optimizer | Adam |
| Optimizer: learning rate | 0.0001 |
| Optimizer: $\beta_1$ | 0.9 |
| Optimizer: $\beta_2$ | 0.999 |
| Optimizer: $\epsilon$ | 0.00015 |
| Max gradient norm | 10 |
| Priority exponent | 0.5 |
| Priority correction | $0.4 \rightarrow 1$ |
| Exploration | Noisy nets |
| Noisy nets parameter | 0.5 |
| Training steps | 100K |
| Evaluation trajectories | 100 |
| Min replay size for sampling | 2000 |
| Replay period every | 1 step |
| Updates per step | Variable (1, 2, 4, 8, 16) |
| Multi-step return length | 10 |
| Q network: channels | 32, 64, 64 |
| Q network: filter size | $8 \times 8, 4 \times 4, 3 \times 3$ |
| Q network: stride | 4, 2, 1 |
| Q network: hidden units | 512 |
| Non-linearity | ReLU |
| Target network update period | 1 |
| $\lambda$ (SPR loss coefficient) | 2 |
| $K$ (SPR prediction depth) | 5 |
| Data Augmentation | Shifts ($\pm 4$ pixels) Intensity(scale=0.05) |
| $\tau$ (EMA coefficient) | 0.995 |
| Reset Interval (gradient steps) | 40,000 |
| Layers getting hard reset | Final 2 |
| Shrink and Perturb $\alpha$ | 0.8 |
| Action selection | Target network |

| Parameter | Setting |
|---|---|
| Discount factor | 0.99 |
| Minibatch size | 256 |
| Optimizer (all) | Adam |
| Optimizer (all): learning rate | 0.0003 |
| Optimizer (all): $\beta_1$ | 0.9 |
| Optimizer (all): $\beta_2$ | 0.999 |
| Optimizer (all): $\epsilon$ | 0.00015 |
| Networks (all): activation | ReLU |
| Networks (all): n. hidden layers | 2 |
| Networks (all): hidden units | 256 |
| Initial Temperature | 1 |
| Replay Buffer Size | $10^6$ |
| Updates per step | Variable (1 to 128) |
| Target network update period | 1 |
| $\tau$ (EMA coefficient) | 0.995 |
| Reset Interval (gradient steps) | 2560000 |
| Layers getting hard reset | All |

Table 5: Hyperparameters for SR-SPR and SR-SAC. The ones introduced by this work are at the bottom of the respective tables.

| Environment | Tasks |
|---|---|
| walker | run |
| quadruped | run, walk |
| reacher | hard |
| humanoid | run, walk, stand |
| swimmer | swimmer6 |
| cheetah | run |
| hopper | hop, stand |
| acrobot | swingup |
| pendulum | swingup |
| finger | turn_hard |
| fish | swim |

Table 6: The tasks from the DMC15 benchmark. We chose commonly-employed DMC tasks for which the optimal policy is not immediately found by SAC according to https://github.com/denisyarats/pytorch_sac#results.

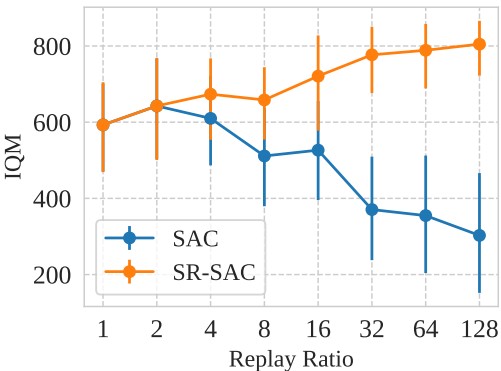

Figure 16: Scaling curve for SR-SAC and SAC on DMC15-1M.

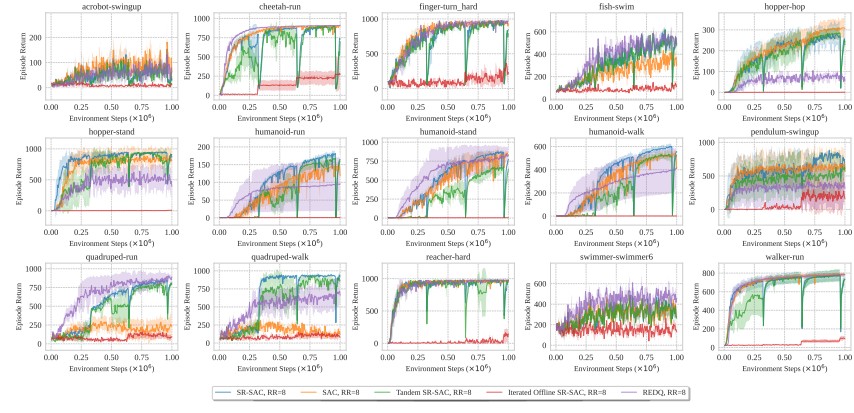

Figure 17: Evaluation Returns on individual DMC15 environments for replay ratio 8.

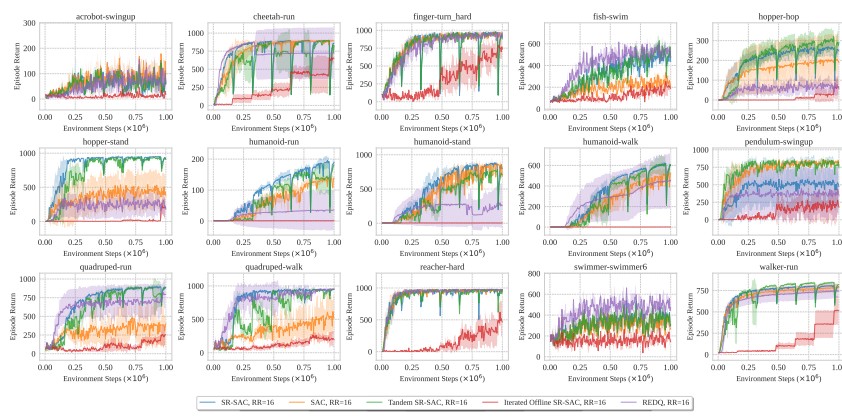

Figure 18: Evaluation Returns on individual DMC15 environments for replay ratio 16.

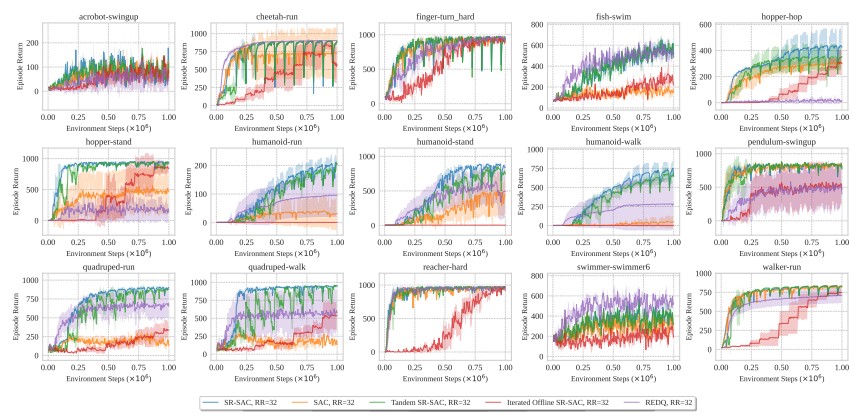

Figure 19: Evaluation Returns on individual DMC15 environments for replay ratio 32.

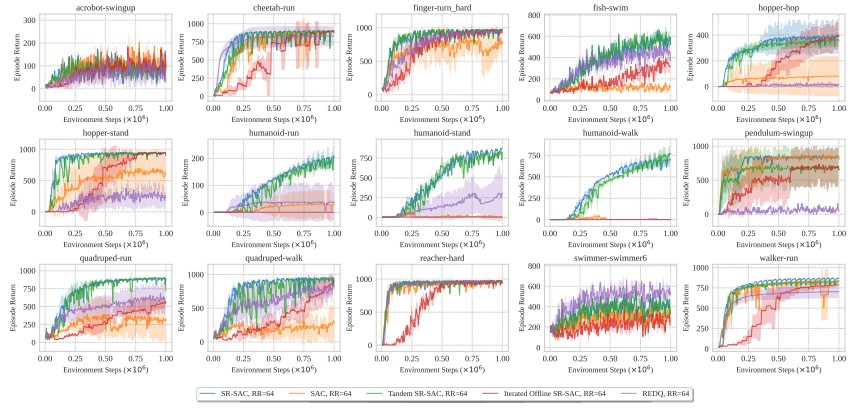

Figure 20: Evaluation Returns on individual DMC15 environments for replay ratio 64.

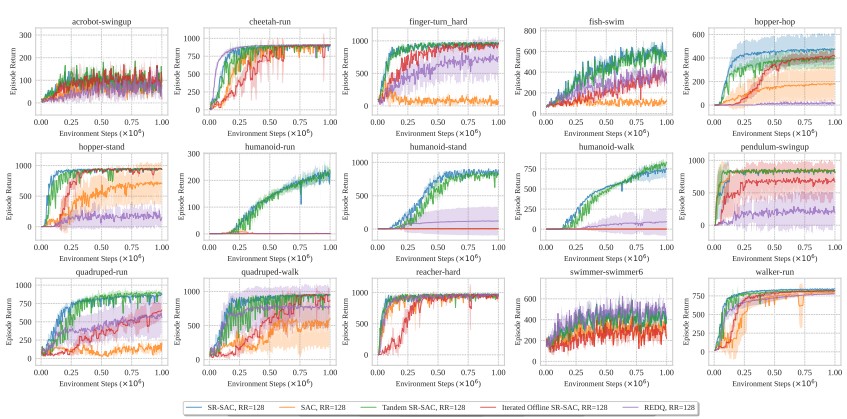

Figure 21: Evaluation Returns on individual DMC15 environments for replay ratio 128.

