# OpenReview forum: "Sample-Efficient Reinforcement Learning by Breaking the Replay Ratio Barrier"
_ICLR.cc/2023/Conference — ICLR 2023 notable top 5%_

### Official Review · Reviewer_X4J7 · 2022-10-17

**Confidence:** 5
**Correctness:** 4
**Technical Novelty And Significance:** 3
**Empirical Novelty And Significance:** 3
**Recommendation:** 8

**Clarity, Quality, Novelty And Reproducibility:**

The paper is clear and generally high quality. Reproducibility is likely somewhat easy, but authors don't list hyperparameters. The work is original, though it is somewhat of an extension of prior work.

**Strength And Weaknesses:**

## Strengths

**Clarity:** The paper is written clearly and generally easy to understand.

********************************************Framing/Contribution:******************************************** The paper’s contribution is clear to understand.

**Experiments/Results:** The authors perform extensive experiments that are convincing of the usefulness of high replay ratios when combined with periodically fully/partially resetting network parameters.

## Weaknesses

**Clarity:**

- In the related works section, first paragraph, many italicized terms are used but not explained. I was familiar with all terms except ************primacy bias************, which I had to look up separately while reading. But in general, these terms should be quickly defined to make the paper easier to read.
- The first paragraph of 5.1.2 is written poorly and hard to read, please fix it.

**Experiments:**

- IQMs are listed (which is great) but the authors should probably explain IQM and the “Fraction of runs with score > $\tau$” statistics for readers unfamiliar with the benchmarking paper that introduced these metrics.
- While the provided analysis on different algorithm design choices in light of high replay ratios is useful, I’m not sure that what the authors analyzed is the most sensible thing to analyze. The paper is about replay ratio scaling through resetting network parameters. Therefore, there should be analysis on two things: 1) different replay ratios, and 2) how/when to reset network parameters. The authors provide analysis of 1) through trying different replay ratios, but insufficient analysis of 2) as in Section 4 they define their reset strategies and just stick with it. I think the paper really should have this analysis to be more useful as an empirical study to inform design choices for other researchers.
- In combination with the experiments in Section 5, it would be useful to have an experiment with an offline RL algorithm (perhaps IQL or CQL). Offline RL is increasingly popular and also being deployed on real robots, so the authors should study their replay ratio increases with an offline algorithm to make the paper more useful for researchers.

****************************Minor details:****************************

- “The number of agent updates per environment step is usually called ************replay ratio************, and most standard algorithms were designed to have values around or below 1” → not sure if they were ********designed******** to have values around or below 1, I think this statement would be more accurate if it was something like “most standard algorithms are trained with a replay ratio around 1.”
- “under tasks switches” → “under task switches”
- “with a same algorithm” → “with the same algorithm”
- “for evaluation and comparisons, we follow the protocol suggested by  Agarwal et a. (2021)” missing a period

**Summary Of The Paper:**

The authors empirically study how combining replay ratio increases with network parameter resets can greatly improve sample efficiency of off-policy methods. Their experiments demonstrate superior performance on existing benchmarks with simple modifications to existing methods.

**Summary Of The Review:**

I think this paper is useful and should be accepted to this conference, however, I have some suggestions for improvements with the experiments. I do think that the authors are missing one key set of analysis, which is why I am recommending a weak accept.

---

> ### Author Response · Authors · 2022-11-14
> **Response to Reviewer X4J7**
>
> Thank you for your feedback!
>
> > But in general, these terms should be quickly defined to make the paper easier to read.
>
> Thank you, this is a good suggestion. We have added a table of definitions and references to simplify this (see Table 3 in the appendix).
>
> > Therefore, there should be analysis on …  how/when to reset network parameters
>
> We have added a study on the impact of when to reset parameters (the reset interval) for both SR-SPR and SR-SAC (Figure 9), illustrating that performance is reasonably robust to this parameter. As for the question of how to reset parameters, Figure 7 contains two ablations examining this for SR-SPR – one where only hard resets are used (-SP) and one where only soft resets are used (SP only). These demonstrate that our mixed hard-soft approach is superior by a large margin to a soft-only approach, and by a smaller margin to a hard-only approach.
>
> > IQM and the “Fraction of runs with score > τ” statistics for readers unfamiliar with the benchmarking paper that introduced these metrics.
>
> We added more precise explanations for these terms in the updated version of the paper.
>
> > In combination with the experiments in Section 5, it would be useful to have an experiment with an offline RL algorithm (perhaps IQL or CQL).
>
> We have added an experiment to the appendix considering the use of an offline RL algorithm, IQL, in online RL. In this case we find that IQL performed poorly with and without resets, likely due to its impact on reducing exploration (see Section B.1 in Appendix for more detailed comments).
>
> In an offline setting, we do not necessarily expect our method to yield large performance improvements. The core element in our method, particularly as applied in SR-SAC, is that the replay buffer increases in quality throughout training, meaning that the agent starts from a better position after each reset. In offline RL this is not the case, as the replay buffer is fixed, meaning that the result of training in each iteration would be expected to be identical.
> While we think it is possible that iteratively applying soft resets might improve performance by improving representations, as for example Zhou et al (2021) demonstrated in supervised learning, we leave this as a question for future work.
>
> > authors don't list hyperparameters
>
> We added detailed hyperparameter lists for both the discrete and continuous control settings in Table 5 in the appendix.
>
> **References**
>
> Zhou, H., Vani, A., Larochelle, H. and Courville, A., 2021, September. Fortuitous Forgetting in Connectionist Networks. In International Conference on Learning Representations.

---

> > ### Comment · Reviewer_X4J7 · 2022-11-14
> > **Response to authors**
> >
> > Thanks for the comprehensive response. I believe you have addressed my major concerns; I have just one more comment:
> >
> >
> > > In an offline setting, we do not necessarily expect our method to yield large performance improvements. The core element in our method, particularly as applied in SR-SAC, is that the replay buffer increases in quality throughout training, meaning that the agent starts from a better position after each reset. In offline RL this is not the case, as the replay buffer is fixed, meaning that the result of training in each iteration would be expected to be identical. While we think it is possible that iteratively applying soft resets might improve performance by improving representations, as for example Zhou et al (2021) demonstrated in supervised learning, we leave this as a question for future work.
> >
> > This is my fault for miscommunicating my original statement, but what I meant was more of the offline RL + online fine-tuning scenario in which the agent is presented with more data online and therefore the replay ratio should ideally make a difference. This offline to online fine-tuning procedure will likely be important for real-world RL applications and there is a potential for "SR-IQL" or something of that sort to be effective here.
> >
> >
> > Otherwise, thanks for addressing my concerns. I will take some time to thoroughly review the other reviews and author responses and consider changing my score.

---

> > > ### Author Response · Authors · 2022-11-18
> > > **Experiment in the offline+fine-tuning setting**
> > >
> > > Thank you for your answer and for the clarification! We are glad that we addressed your major concerns.
> > >
> > > We believe generalizing the approach presented in the paper to the fine tuning setting is indeed a very interesting research direction. To showcase this, we implemented the experiment you have described, training an algorithm that can be called SR-IQL on two of the antmaze tasks from the D4RL benchmark, with 1 million steps for fine tuning after 1 million steps of offline pretraining. During fine tuning, we use a replay ratio of 10 and a reset interval of 2 million update steps.
> > >
> > > Results in Figure 12 of the updated version of the paper show that SR-IQL clearly outperforms IQL on one of the two tasks. We are excited about the possibility of future work extending our methodology and analyses to this setting, which will perhaps benefit from specialized solutions, and thank you again for your insightful suggestion about this experiment. We hope this addresses your comment.

---

> > > > ### Comment · Reviewer_X4J7 · 2022-11-18
> > > > **Great response**
> > > >
> > > > This is great and I did not expect this additional experiment. I have also looked at the whole paper again in addition to other reviews. I agree with all other reviewers: this is a clear accept. I am updating my score.

---

### Official Review · Reviewer_MFRL · 2022-10-21

**Confidence:** 4
**Correctness:** 3
**Technical Novelty And Significance:** 3
**Empirical Novelty And Significance:** 3
**Recommendation:** 8

**Clarity, Quality, Novelty And Reproducibility:**

See my comments in Strengths and Weaknesses.


**Strength And Weaknesses:**

**Strengths:**
[Clarity & Quality] The paper is basically well-written and easy to read.
[Novelty] Analyses to better understand the reset method [1] are conducted.
 * In the original reset paper [1], the reset method was not compared to methods with a sample efficient method, such as REDQ. The experiments in the paper under review show that the reset method is generally more efficient than the existing sample-efficient method.
 * Experiments in tandem and iterated offline settings are conducted.  In addition, the effect of combining online updates with offline updates is evaluated. This combination approach addresses the temporal performance degradation of resetting, which was one of the main limitations of the reset method.
 * Implementation decisions for discrete control problems and the trade-off between computational resources and performance are also analysed.
 * The experiment results and findings should be beneficial especially to practitioners. Also, as a reset method for RL is a relatively new one and experiments about it have not been stacked so far, the experiment results provided in the  paper under review will be beneficial for a future advance of the reset method.)

**Weaknesses**
[Novelty] Not so much progress has been made from [1].
 * There is only a few technical difference from the original reset method: for continuous control, the proposed reset method is based on "updates" steps rather than "environment steps", and for discrete control, the Shrink and perturb style encoder update is used.
 * As with [1], the reset interval is treated as a hyperparameter,
and it is still unclear the optimal timing for reset (i.e., we still need hyper-parameter tuning for it).

[Clarity ] It is unclear how much performance is sensitive to the hyperparameter of the reset interval.
 * For experiments for both discrete and continuous environments, reset intervals are set to magic numbers. No detailed discussion about how this hyperparameter affects overall performance is provided. This makes understanding how the insights observed in the experiments are generalizable a bit difficult.



**Minor comments**
> Recent approaches in continuous control leveraged high replay ratios as a strategy to improve sample efficiency through the use of ensembles of value functions (Chen et al., 2021) or normalization strategies (Smith et al., 2022).

The following approaches also use ensemble and regularization in high replay ratio settings.

1. Takuya Hiraoka and Takahisa Imagawa and Taisei Hashimoto and Takashi Onishi and Yoshimasa Tsuruoka, Dropout Q-Functions for Doubly Efficient Reinforcement Learning, International Conference on Learning Representations, 2022
2. Wu, Yanqiu and Chen, Xinyue and Wang, Che and Zhang, Yiming and Zhou, Zijian and Ross, Keith W, Aggressive Q-Learning with Ensembles: Achieving Both High Sample Efficiency and High Asymptotic Performance, arXiv preprint arXiv:2111.09159, 2021

> To answer this question, we resort to what we call iterated offline RL, ...

Is this the same setting as the following paper?

Matsushima, Tatsuya and Furuta, Hiroki and Matsuo, Yutaka and Nachum, Ofir and Gu, Shixiang, Deployment-efficient reinforcement learning via model-based offline optimization, arXiv preprint arXiv:2006.03647, 2020


> Replay Ratio Scaling: Change in an agent’s performance caused by doing more updates for a fixed number of environment interactions.

Regarding SR-SAC, does replay ratio here mean, critic update ratio, actor update ratio, or both?  In the REDQ paper, replay (UTD) ratio means critic update ratio (actor is updated only once per environment step).


> The progressive loss of ability to learn and generalize of neural networks and its interaction with RL was only one of the recent empirical discoveries about deep RL.

This sounds like an exaggeration. Why "only" one?


**Summary Of The Paper:**

Overall, the paper provides deeper analyses of the reset method proposed by Nikishin et al. [1].
The paper under review shows that resetting the agent after every certain number of updates prevents performance degradation even at large replay ratio settings.
In addition, several experiments and analyses that had not been done in [1] were made:
1. comparison with existing sample-efficient methods such as REDQ, 2. evaluation of the reset method in iterated offline and tandem settings, 3. analysis of implementation-level design decision in discrete control settings, and 4. evaluation of trade-offs between computational resources and performance.

[1] Nikishin, Evgenii and Schwarzer, Max and D'Oro, Pierluca and Bacon, Pierre-Luc and Courville, Aaron, The Primacy Bias in Deep Reinforcement Learning, Proceedings of the 39th International Conference on Machine Learning, 2022


**Summary Of The Review:**

Overall, I lean somewhat toward Accept.
The main weakness of the paper is the lack of novelty, i.e., the methods are not very different from the existing one, and the reset interval is still treated as a hyperparameter (and the analysis of how much it affects the results is not provided in the paper under review).
On the other hand, the various experiments conducted to gain a deeper understanding of the reset method are useful and this is the main strength of the paper.
I think that the results/insights from these experiments will accelerate the development of the reset method in the future.

---

> ### Author Response · Authors · 2022-11-14
> **Response to Reviewer MFRL**
>
> Thank you for your feedback!
>
> > For experiments for both discrete and continuous environments, reset intervals are set to magic numbers. No detailed discussion about how this hyperparameter affects overall performance is provided. This makes understanding how the insights observed in the experiments are generalizable a bit difficult.
>
> We have added evidence (Figure 9 in Appendix) demonstrating that positive replay ratio scaling occurs across a wide range of reset intervals. While certain reset intervals are roughly optimal, overall performance is not excessively sensitive to this choice of parameter, unless a too large reset frequency implies the total lack of resets.
>
> > The following approaches also use ensemble and regularization in high replay ratio settings.
>
> Thank you for pointing these out! We have now included them as related work.
>
> > The progressive loss of the ability to learn and generalize of neural networks and its interaction with RL was only one of the recent empirical discoveries about deep RL. This sounds like an exaggeration. Why "only" one?”
>
> We merely meant to say that there has been a great deal of empirical research in DRL recently, and that many other empirical discoveries have been made in addition to this. We fixed the wording in the updated version of the paper.
>
> > There is only a few technical difference from the original reset method
>
> We believe one crucial part of the value of our work lies in demonstrating that the original reset method we use as a starting point can open up entirely new training regimes and types of scaling when appropriately modified. This consists both of reparameterizing the reset frequency in terms of updates (in both SR-SAC and SR-SPR) and of proposing a set of modifications to the original reset method that allow replay ratio scaling (in SR-SPR). These modifications are crucial, as we show in our analyses, and in some cases perhaps unintuitive.
>
> > ...Is this the same setting as the following paper?
>
> Yes, it is indeed a similar setting, thank you for the suggestion. Interpreted through their lens, our results indicate that SR-SAC is very deployment-efficient compared to baseline SAC.
>
> > Regarding SR-SAC, does replay ratio here mean, critic update ratio, actor update ratio, or both?
>
> We change the update ratios for actor and critic in parallel, so it refers to both.

---

> > ### Comment · Reviewer_MFRL · 2022-11-15
> > **Response to authors**
> >
> > Thank you for your response.
> > Overall, I am satisfied with the response and improve the score as WA -> A.

---

### Official Review · Reviewer_qf9b · 2022-10-23

**Confidence:** 4
**Correctness:** 4
**Technical Novelty And Significance:** 3
**Empirical Novelty And Significance:** 4
**Recommendation:** 8

**Clarity, Quality, Novelty And Reproducibility:**

**Clarity and Quality.**
- There's a typo in Section 5.2: ` achieving robust replay ratio scaling for SR-SPR is requires due to its shorter training period and more complex function approximation.`
- In Figure 3 and Table 2, it could be more helpful for clarity if baselines are introduced with acronyms along with their references.
- For self-containedness, it would be nice to include the formulation or more detailed explanation on SPR

**Novelty.**
- Interesting and novel empirical observations

**Reproducibility.**
- It seems that the approach is not difficult to reproduce; but it would be nice to include source codes for this

**Strength And Weaknesses:**

Strengths
- Well written, intuitive approach
- Strong performance with a very simple idea
- Exhaustive experiments and interesting analysis and discussions

Weaknesses
- Given the recent surge of leveraging pre-trained representations for vision-based RL, it would be an interesting investigation to see what would happen when considering such a pre-trained perception module.
- Investigating the role of replay ratio scaling for model-based approaches, especially the effect on the world models and corresponding policies obtained from the models, would be an interesting addition to the paper, but I don't think this is a necessary and could be a future work.

**Summary Of The Paper:**

This paper presents that we can significantly improve the sample-efficient of prior deep RL approaches by increasing the number of updates per environment steps, and shows that resetting all parameters or part of parameters is critical. The paper shows improved performance on a variety of benchmark tasks, and provides interesting analysis and observations that can be related to the offline RL literature.

**Summary Of The Review:**

The paper consists of an intuitive approach, nice explanation, strong performance, well supported claims, and interesting discussions. I recommend the paper to be accepted and has no major concerns.

---

> ### Author Response · Authors · 2022-11-14
> **Response to Reviewer qf9b**
>
> Thank you for your feedback!
>
> > it would be an interesting investigation to see what would happen when considering such a pre-trained perception module.
>
> We appreciate the suggestion and find the proposed investigation interesting. We are in the process of implementing SR-SPR on top of a pretraining scheme; unfortunately, because of interoperability problems of our codebase with existing methods, it is not possible to easily run this experiment on top of previous representation learning methods in the short timeframe allowed by the rebuttal period.
>
> > Investigating the role of replay ratio scaling for model-based approaches, especially the effect on the world models and corresponding policies obtained from the models, would be an interesting addition to the paper, but I don't think this is a necessary and could be a future work.
>
> We believe the study of replay ratio scaling and resets in the model-based reinforcement learning setting is indeed particularly interesting. This work was devoted to model-free methods, but future work can investigate scaling and resets in model and policy updates. We will post any preliminary findings we might have before the end of the rebuttal period.
>
> > For self-containedness, it would be nice to include the formulation or more detailed explanation on SPR
>
> We have expanded the explanation of SPR given in our experimental section as much as space allows.
>
> > In Figure 3 and Table 2, it could be more helpful for clarity if baselines are introduced with acronyms along with their references.
>
>
> We added more precise reference to corresponding work in the paper.
>
> > it would be nice to include source codes for this
>
> We plan to release open-source code when we publicly announce the paper.

---

> > ### Comment · Reviewer_qf9b · 2022-11-18
> > **Response**
> >
> > Thanks for the detailed response! Additional results that fine-tune the pre-trained representations also look nice. I'm not updating the score but leaving the response to notice that I have the rebuttal response from the authors.

---

> ### Author Response · Authors · 2022-11-15
> **Results on finetuning pretrained representations**
>
> > Given the recent surge of leveraging pre-trained representations for vision-based RL, it would be an interesting investigation to see what would happen when considering such a pre-trained perception module.
>
> We conducted an experiment (B.2 in the appendix of the current version of the paper) where we finetuned pretrained encoders using SPR and SR-SPR. We find that both SPR and SR-SPR benefit from being initialized with pretrained encoders at all replay ratios relative to training from scratch. However, unlike SPR, SR-SPR is able to combine the benefits of pretraining and replay ratio scaling to achieve much higher performance during finetuning.

---

### Author Response · Authors · 2022-11-14
**General Response**

We thank the reviewers for their helpful comments and feedback. Overall, reviewers appreciated our analyses and empirical results, finding our observations interesting and novel (Reviewer qf9b), our findings beneficial to practitioners (Reviewer MFRL) and our experiments convincing of the usefulness of high replay ratios when combined with resets (Reviewer X4J7). They also unanimously appreciated the writing quality of the paper.

Two reviewers requested additional details on the impact of the precise reset strategy chosen, particularly the reset interval. We have provided additional experimental results illustrating the impact of varying the reset frequency for SR-SAC and SR-SPR (Fig. 9 in the new revision), demonstrating that a wide range of reset frequencies may all enable positive replay ratio scaling.

Reviewers also raised a range of individual issues, which we have addressed in the new revision. This includes clarifying the text, adding extra experimental details, and improving our discussion of related work. We address these in replies to each reviewer below.

---

### Decision · Program_Chairs · 2023-01-20

**Decision:**

Accept: notable-top-5%

**Justification For Why Not Higher Score:**

N/A

**Justification For Why Not Lower Score:**

I cannot come up with any reasonable weaknesses for the paper. The authors have some very nice insights on sample efficient deep reinforcement learning that are going to be interesting for both RL practitioners and more theoretically-minded researchers.

**Metareview: Summary, Strengths And Weaknesses:**

(a) Summary
The paper builds on the reset method proposed by Nikishin et al. and replay ratio scaling. The paper demonstrates the benefits of these ideas and how they enable designing new sample efficient learning settings. The approaches are very extensively evaluated and compared with experiments.

(b) Strength
- The paper is well written and easy to understand
- The contributions are clearly described
- The proposed methods are elegant and practically relevant
- The proposed modifications compared to the original methods are shown to be crucial
- Design decisions are evaluated
- The observations and discussions are interesting
- The experimental results are very thorough and convincing

(c) Weaknesses
- The writing issues have all been fixed
- The paper does not provide any fundamentally novel methods/components nor deep theoretical analysis/proofs - but that's also not the point of the paper.

**Note From Pc:**

if the above contains the word "oral" or "spotlight" please see: "oral" presentation means -> notable-top-5% and "spotlight" means -> notable-top-25%. As stated in our emails, we are disassociating presentation type from AC recommendations

**Summary Of Ac-Reviewer Meeting:**

N/A